# Multimodal confined water dynamics in reverse osmosis polyamide membranes

Fabrizia Foglia (ID) [1,3✉], Bernhard Frick (ID) [2], Manuela Nania[1], Andrew G. Livingston (ID) [1,4] & João T. Cabral (ID) [1✉]

While polyamide (PA) membranes are widespread in water purification and desalination by reverse osmosis, a molecular-level understanding of the dynamics of both confined water and polymer matrix remains elusive. Despite the dense hierarchical structure of PA membranes formed by interfacial polymerization, previous studies suggest that water diffusion remains largely *unchanged* with respect to bulk water. Here, we employ neutron spectroscopy to investigate PA membranes under precise hydration conditions, and a series of isotopic contrasts, to elucidate water transport and polymer relaxation, spanning ps-ns timescales, and Å-nm lengthscales. We experimentally resolve, for the first time, the multimodal diffusive nature of water in PA membranes: in addition to (slowed down) translational jump-diffusion, we observe a long-range and a localized mode, whose geometry and timescales we quantify. The PA matrix is also found to exhibit rotational relaxations commensurate with the nanoscale confinement observed in water diffusion. This comprehensive 'diffusion map' can anchor molecular and nanoscale simulations, and enable the predictive design of PA membranes with tuneable performance.

[1] Department of Chemical Engineering, Imperial College London, London SW7 2AZ, UK. [2] Institut Laue Langevin, 71 avenue des Martyrs - CS 20156 - 38042, Grenoble CEDEX 9, France. [3] Present address: Department of Chemistry, Christopher Ingold Laboratory, University College London, London WC1H 0AJ, UK. [4] Present address: School of Engineering and Materials Science, Queen Mary University of London, Mile End Road, London E1 4NS, UK. ✉email: f.foglia@ucl.ac.uk; j.cabral@imperial.ac.uk

The scarcity of "fresh" water, amounting to less that 1% of the Earth's water supply[1], is one the most pressing societal challenges of our time, and is expected to grow as water needs for human consumption, agriculture and industry increase. Polymeric membranes are extensively used in separation and purification processes, including water desalination[2–4]. Reverse osmosis (RO) membrane processes, in particular employing thin-film composite (TFC) aromatic polyamide (PA) membranes, offer high-energy efficiency as well as comparatively high selectivity and permeability and play a key role in addressing this challenge[1]. The active layer of these membranes is generally a rough and crumpled, highly cross-linked PA film, formed by the interfacial polymerization (IP) at the aqueous/organic interface between an aromatic diamine (m-phenylenediamine, MPD) and trimesoyl chloride (TMC), with an overall thickness of a few hundred nm, supported by a thick, porous, non-selective layer. Despite the widespread use of RO, the discovery and engineering of novel materials[5,6] as well as process design and efficiency optimization[1,6,7], further improving separation performance has remained non-trivial. In part, this is due to a surprising lack of a molecular-level understanding of the water dynamics confined within the membrane, of the PA matrix relaxation, and their possible coupling, which we seek to resolve.

Transport through the active layer is generally described by solution-diffusion macroscale models[2,8], which assume that the active layer is effectively non-porous and that water and salt partition into the membrane, diffuse along the chemical potential gradient and eventually desorb into the permeate. Such descriptive, engineering models are then parameterized by experimentally measured constants, enabling different membranes, preparation methods, and conditioning processes to be compared. The importance of the active layer nanostructure (or "geometry") and intrinsic water transport within the polymer matrix is, however, widely recognized[9] and has been incorporated in hybrid coarse-grained PA models.

PA membrane formation via IP proceeds dynamically via a mechanism of cluster formation, diffusion-limited aggregation, and percolation, resulting in a well-known *inhomogeneous* film structure with a distribution of pore sizes, functional group asymmetry and surface polarization[10–18]. TMC/MPD oligomeric "clusters" are thought to form and coalesce to yield a "coherent film" of dense clusters within lower-density structures, which have been analytically and numerically modeled[14,15,19,20]. Such "aggregate" and "network" pores, as well as "lagoons", are thus expected to impose complex water transport processes, modulated by these regions of varying polymer density within the membrane[11–13,21–24].

Molecular dynamics (MD) simulations of water diffusion within PA membranes[10–13,22,25–28], generally find diffusion coefficients approximately one order of magnitude lower than that of bulk water[29,30]. Kotelyanskii et al.[25] reported two values for water diffusivity, $D \simeq 0.2$ and $0.7 \times 10^{-5}$ cm$^2$ s$^{-1}$ at constant volume and pressure, at 300 K, with a (translational) jump length of ~3 Å. More recent simulations observe a heterogeneous distribution of water diffusivity[11,22,27], often described by a bimodal distribution, whose physical interpretation generally invokes dynamics within distinct pore types or interfacial vs. confined water. Ding et al.[22] find $D \simeq 0.2$ and $0.6 \times 10^{-5}$ cm$^2$ s$^{-1}$ for confined water and in the "interfacial region", respectively, in broad agreement with recent work[28]; in addition, MD studies generally recover bulk dynamics in the external water "reservoirs", as expected. Hughes and Gale[13] propose that water within the membrane is arranged in pockets connected by "chains" of water molecules, while Kolev and Freger[12] find strong next-neighbor correlations and random long-distance water–water correlations, associated with the presence of small and large

pores. Positron annihilation lifetime spectroscopy (PALS) measurements infer pore sizes $\simeq 2.1$–$2.4$ and $3.5$–$4.5$ Å, interpreted as due to network and aggregate pores, respectively[21], in line with other membranes[31], and in broad agreement with simulation[11,12,22,27] and small-angle X-ray scattering[32].

Surprisingly, previous experimental investigations of water dynamics within PA membranes[33–36] have found translational diffusion coefficients only slightly reduced ($\simeq 1.90$ and $2.06 \times 10^{-5}$ cm$^2$ s$^{-1}$ at 300 K) with respect to those of pure water, and no evidence of spatial confinement or bimodal diffusion mechanisms. We, therefore, seek to elucidate the nature of water dynamics with PA membranes, how its mechanism and kinetics are affected by confinement, and the relation between water diffusion and the heterogeneous PA nanostructure. We employ neutron time-of-flight (TOF) and backscattering (BS) spectroscopy to examine water diffusion and PA membrane dynamics, which we decouple by selective deuteration of water (H$_2$O and D$_2$O), and systematically investigate hydration level (0% and 100% relative humidity) and hydration procedure to resolve this conundrum. Further, differential scanning calorimetry (DSC) was carried out to discriminate between confined and "bulk" water, and our dynamics data were benchmarked with MD simulations, and macroscopic performance data. Informed by the spatial heterogeneity of PA membranes and simulation of water dynamics, it seems reasonable to expect water to exhibit a space-dependent distribution of diffusion coefficients. Our hypothesis is that water diffusion through the network (or "bulk" polymer matrix) and aggregate "pores" can be experimentally resolved with a judicious choice of complementary instrumental resolutions. Other physical pictures can, however, also be conceived, for instance in terms of interfacial and bulk water diffusion, sub-diffusive dynamics within the dense polymeric membrane, and possible spatio-temporal coupling with multiscale relaxation processes within the polymer network.

## Results and discussion

Incoherent Quasi-Elastic Neutron Scattering (QENS) provides a direct measurement of the self-correlation function of *all* hydrogen atoms, whose incoherent cross-section dominates the signal, within a sample. In order to separately resolve the polymer dynamics and water-diffusion signals, our experiment and analysis were decoupled in stages. Employing dry PA membranes and membranes hydrated with D$_2$O, we first resolve the segmental dynamics of the dry and hydrated polymer alone; equipped with this knowledge, we can then isolate the water signal in PA membranes hydrated in H$_2$O, and experimentally resolve the confined water dynamics. Significantly, in this work, we investigate two hydration procedures in both H$_2$O and D$_2$O contrasts: PA membranes were hydrated in the vapor phase, up to 100% hydration as measured by water uptake, as well as by direct immersion in water followed by careful drying of excess surface water, as customarily done experimentally (including in previous QENS studies[33,34]).

Hydrogen dynamics are decomposed into atomic vibrations, translations and rotations and, assuming these are uncorrelated, the measured scattering function becomes a convolution of these terms: $S_{inc}(Q,\omega) = S^V \otimes S^T \otimes S^R$[37]. For isotropic, harmonic vibrations this simplifies to $S_{inc}(Q,\omega) = e^{-\frac{1}{3}Q^2\langle u^2 \rangle}\left(S^T \otimes S^R\right)$, and the mean-square displacement $\langle u^2 \rangle$ can be measured directly by fixed window scans at sufficiently low temperatures (below the activation of translational and rotational motions, beyond which it becomes an "apparent" $\langle u^2 \rangle_{apparent}$). The temperature-dependent $\langle u^2 \rangle(T)$ quantifies the spatial extent of delocalization of H atoms, whose scattering cross-section dominates the signal.

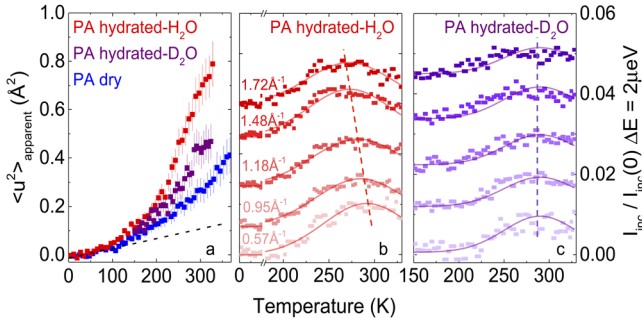

**Fig. 1 Mean-square displacement and IFWS of dry and hydrated PA membranes. a** Temperature dependence of the apparent mean-square displacement $\langle u^2 \rangle_{apparent}$ for dry PA (blue), vapor hydrated in $D_2O$ (purple) and $H_2O$ (red); the dashed line indicates the mean-square displacement, obtained from the Debye–Waller factor. The error bars correspond to the maximum intervals in the low-$Q$ extrapolation. **b** Inelastic fixed window scan (IFWS) for PA hydrated in $H_2O$ and (**c**) hydrated in $D_2O$, at representative $Q$-values; the profiles were vertically shifted for clarity and lines are guides to the eye. Source data files provided.

Dry and hydrated PA films were first analyzed by elastic (EFWS, $\Delta E = 0\,\mu eV$) and inelastic (IFWS, $\Delta E = 2\,\mu eV$) fixed window scans, as a function of temperature (2–380 K) using the high-resolution backscattering spectrometer IN16B. The upper temperature was chosen to exceed typical RO operating conditions (up to 318 K), yet below the glass transition temperature $T_g$ of PA ($\simeq 390\,K$, Supplementary Note 1). The temperature dependence of $\langle u^2 \rangle$ was computed from the EFWS as shown in Fig. 1a, and its low temperature ($\lesssim 100\,K$) slope yields $\langle u^2 \rangle / dT = 0.6 \pm 0.1 \times 10^{-3}\,Å^2\,K^{-1}$ for dry PA, increasing only marginally upon hydration (further detailed in Supplementary Notes 2 and 3). At higher temperatures (300 K for the dry PA sample), the $\langle u^2 \rangle_{apparent}$ reads 0.27 Å², in agreement with MD simulations (~0.3 Å², Supplementary Fig. 3) for dry membranes and increases considerably to 0.44 Å² upon $D_2O$ hydration, and to 0.67 Å², upon $H_2O$-hydration with the additional signal from water. This increase is expected from the observed membrane swelling of $\simeq 15\%$ upon vapor hydration[23]. The corresponding normalized IFWS obtained at $\Delta E = 2\,\mu eV$ are shown in Fig. 1b and c for the PA membrane hydrated with $H_2O$ and $D_2O$. In these scans, the inelastic scattering intensity increases with temperature and reaches a maximum as the QENS broadening matches a given offset energy $\omega_{off}$[38], yielding the temperature range for which the dynamics of the sample reaches relaxation times $\tau$ corresponding to the energy offset. Diffusive and local (e.g., rotational) motions can be discriminated by the $Q$-dependence or -independence, respectively, of the characteristic maximum. Significantly, these results demonstrate that data from PA membranes hydrated with $H_2O$ yield information from $H_2O$ diffusion, while those from PA hydrated with $D_2O$ isolate the polymer dynamics in the hydrated state, permitting their decoupling in the scattering data.

**Polyamide matrix dynamics: the dry state.** QENS measurements of dry PA membranes, shown in Fig. 2a and b obtained by TOF and BS at two energy resolutions, are interpreted in terms of local rotational relaxations, as translational motions are absent ($S^{tr} = \delta(\omega)$) in the cross-linked sub-$T_g$ PA network. Since not all H atoms in the polymer are mobile at a given temperature, we define a fraction of mobile protons $\phi_P^m$. The immobile fraction, $1 - \phi_P^m$, therefore gives rise to an elastic scattering term $\delta(\omega)$ within the measurement time window.

Experimentally, we identify two rotational relaxation processes: a faster, small amplitude $SA$ relaxation (predominantly visible by TOF), and a slower, large-amplitude relaxation $LA$ (predominantly visible by BS). A combination of TOF and BS experiments, probing complementary timescales, is able to resolve both relaxations. We, therefore, write the scattering law describing the polymer $P$ signal as:

$$S_P(Q,\omega) = (1 - \phi_P^m)\delta(\omega) + \phi_P^m \Big[(1-f)S_{P_{SA}}(Q,\omega) + f\Big(S_{P_{SA}}(Q,\omega) + S_{P_{LA}}(Q,\omega)\Big)\Big]$$ (1)

where a fraction of mobile protons $(1-f)$ undergoes solely $SA$ motions, and fraction $f$ undergoes both $SA$ and $LA$ motions. Equation (1) thus reads

$$S_P(Q,\omega) = (1 - \phi_P^m)\delta(\omega) + \phi_P^m S_{P_{SA}}(Q,\omega) + \phi_P^m f S_{P_{LA}}(Q,\omega)$$ (2)

and each rotational relaxation component is described by an elastic and quasi-elastic term[37]:

$$S_i(Q,\omega) = A_{0,i}(Q)\delta(\omega) + (1 - A_{0,i}(Q))\frac{1}{\pi}\frac{\Gamma_i}{\omega^2 + (\Gamma_i)^2}$$ (3)

where $i$ stands for $P_{LA}$ or $P_{SA}$, and $A_{0,i}$ is the elastic incoherent structure factor (EISF), characteristic of the geometry of the rotational motions, illustrated in Fig. 2c–e. In high-resolution BS, the $S_{P_{SA}}$ Lorentzian, describing the fast proton relaxation, appears broad and approaches an energy-independent (flat) background; by contrast, in TOF, the $S_{P_{LA}}$ Lorentzian, accounting for slower proton relaxations, appears narrow and is effectively merges into the elastic line. The combined energy resolutions employed in BS and TOF thus provide a facile route to decoupling $SA$ and $LA$ measurements, in Eqs. (1)–(3). For both Lorentzian profiles, the half-width at half-maximum $\Gamma$ is found to be $Q$-independent (Fig. 3a, b), and the measured EISFs (Fig. 3c, d) are found to be temperature-independent, as expected for a localized rotation. However, these differ somewhat for $SA$ and $LA$ motions, as discussed next.

We consider possible geometries of motion, compatible with the PA network structure, in Fig. 2c–e. We define a monomer unit as TMC:MPD 1:1.5, following the network stoichiometry, comprising 12 H atoms, of which the expected mobile (5) and immobile (7) H atoms are shown, respectively, in green and red. The associated radii of rotation are thus expected to range from $\simeq 3.8$–5.3 Å. A systematic examination of possible geometries (Supplementary Note 4), yields that an out-of-plane rotation between two sites[37,39], depicted in Fig. 2c–e, describes well the EISF for all data:

$$A_{0,P} = \frac{7}{12} + \frac{5}{12}\big(p_1 p_1 + p_2 p_2 + 2 p_1 p_2 j_0(Qd)\big)$$ (4)

where $j_0$ is a Bessel function of the zeroth-order and $d$ is the distance between two the positions, corresponding to a partial flip on the MPD ring at an angle $\alpha$. For $LA$ motions, we obtain a rotation angle of $\alpha \simeq 40°$ and $d = 5.2 \pm 0.1$ Å, extracted from BS data, probing longer timescales (up to ns); for $SA$ motions, occurring at shorter timescales (ps), well-resolved in TOF, this angle decreases to $\alpha \simeq 15°$, and $d = 2.5 \pm 0.1$ Å. Parameters $p_1$ and $p_2$ denote the occupation probabilities between the two sites, and we find the best agreement with the data for $p_1 = 40\%$ and $p_2 = 60\%$ for both $LA$ and $SA$ motions, indicating a slight preference for one conformation. The presence of high- and low-amplitude motions at different frequencies is corroborated by MD simulations (Supplementary Fig. 6h–j). A relaxation time-scale $\tau_P \equiv \hbar/\Gamma_P$ is associated with each profile, and found to be ~10 ps for $SA$ and 1 ns for $LA$ motions, whose Arrhenius temperature dependence is shown in Fig. 3e. A large population

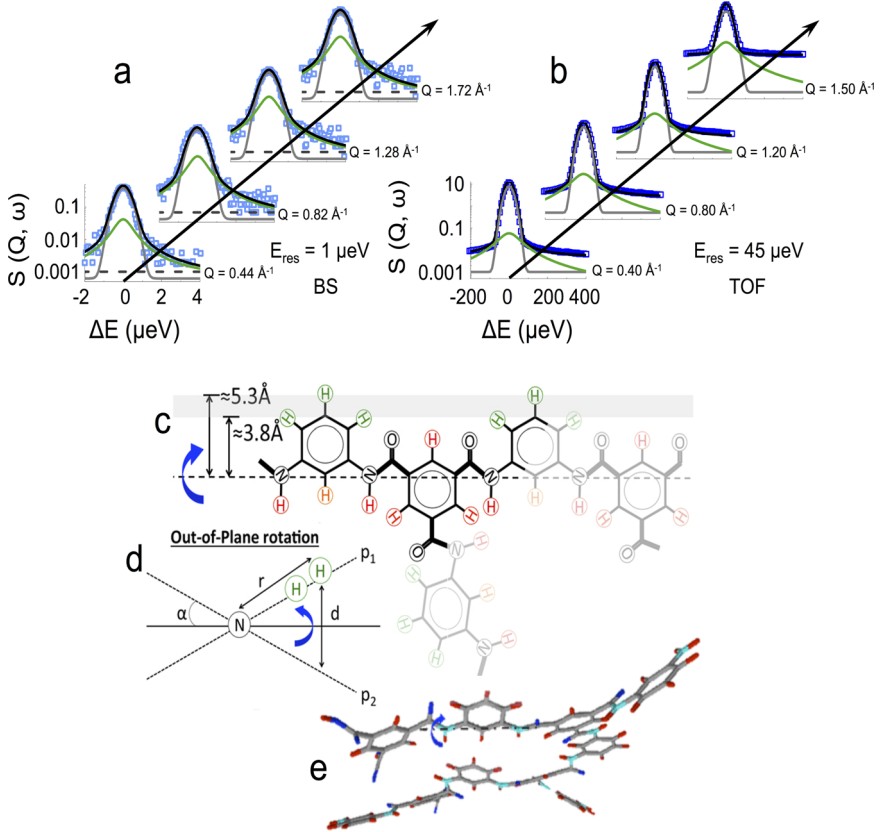

**Fig. 2 QENS data and schematic of dry PA relaxation.** QENS measurements of dry PA at 330 K measured by (**a**) BS, IN16B and (**b**) TOF, IN5 at selected $Q$-values, depicting the $S(Q, \omega)$ data, overall fit (black), instrumental resolution (gray), and Lorentzian profile (green). **c** Schematic of the chemical structure of highly cross-linked PA, labeling H atoms in the TMC-MPD 1:1.5 unit that are expected to be mobile (green) and immobile (red), indicating the proposed rotational axis and distance to mobile protons. **d** Cross-section of the out-of-plane rotation used to compute the EISF[39], and (**e**) corresponding 3D visualization. Source data are available at https://doi.org/10.5291/ILL-DATA.9-11-1809 and https://doi.org/10.5291/ILL-DATA.9-11-5311718.

of polymer segments can be expected to undergo *LA* flips within ns timescales, as compared to *SA* motions within the much narrower ps window, probed by TOF, and we find populations of ~20% and ~80% at 330 K, respectively, for *SA* and *SA-LA* (detailed in Supplementary Notes 4–6), with $\phi_P^m$ and $f$ shown in Fig. 3f–h as a function of temperature.

**Polyamide matrix dynamics: hydrated state**. We next consider QENS results for the PA membrane in a fully hydrated state (Fig. 4). For $D_2O$-hydration (left column), the model above (Eqs. (1)–(3)) holds, as the contribution from water is largely "silent", and the effect of hydration on the polymer alone can thus be examined (Supplementary Note 5). We find that both the geometry and relaxation rate of the motion remain largely unchanged (Fig. 4c, e and Supplementary Figs. 8 and 9), and we associate this observation to the tightly cross-linked nature of PA membranes. A slight (~5%) increase in the population of mobile H atoms, in hydrated membranes with respect to the dry state is observed (Supplementary Fig. 8f–h), and is corroborated by the more pronounced feature in the IFWS for the $D_2O$-hydrated PA with respect to dry PA, at the same temperatures.

**Water dynamics within the membrane**. The $H_2O$-hydrated PA membrane data comprises contributions from both polymer and water dynamics, whose analysis becomes thus complex. Selective deuteration enables us to employ the PA $D_2O$-hydrated data under identical conditions to *fix* the polymer contribution in the analysis, enabling the examination of the $H_2O$ dynamics within the PA membrane (Fig. 4b, d, f, h and Supplementary Note 6).

The model for the scattering law must now consider the proton fraction of both polymer ($\varphi_P$) and water ($\varphi_W$), and the additional signal arising from the water dynamics. In the vapor-hydrated membranes, we find gravimetrically $\varphi_P = 0.67$ and $\varphi_W = 0.33$ (Supplementary Note 7), corresponding to full hydration[23]. We therefore write:

$$S_{inc}(Q, \omega) = e^{-\frac{1}{3}Q^2\langle u^2\rangle}\left(\varphi_P S_P(Q, \omega) + \varphi_W S_W(Q, \omega)\right) \quad (5)$$

where $S_P(Q, \omega)$ is given above Eqs. (1)–(3) and $S_W(Q, \omega)$ is the dynamic structure factor for water. As above, the fraction of mobile water molecules can be expected to vary with temperature, and respond to confinement. Experimentally, combining BS and TOF measurements, we resolve two distinct motions which we assign to translational diffusion (*tr*) and long-range diffusion (*lr*), as detailed in Supplementary Note 6. We assume here that these are decoupled, and associated with the spatial heterogeneous nature of the PA membrane, although other physical pictures are manifestly conceivable. We thus write:

$$S_W(Q, \omega) = (1 - \phi_W^m)\delta(\omega) + \phi_W^m\left[S_W^{str}(Q, \omega) + gS_W^{lr}(Q, \omega)\right] \quad (6)$$

where the first term accounts for the immobile water fraction; $\phi_W^m$ represents the fraction of water molecules undergoing *tr* motions, and $\phi_W^m \times g$ fraction undergoing both *lr* and *tr* motions. Within measurement uncertainty, in TOF data, one Lorentzian profile, with $Q$-dependent width, accounts for the additional water contribution to the signal (as $S_W^{lr}(Q, \omega)$ appears within the elastic line). This is interpreted as due to the translational diffusion of

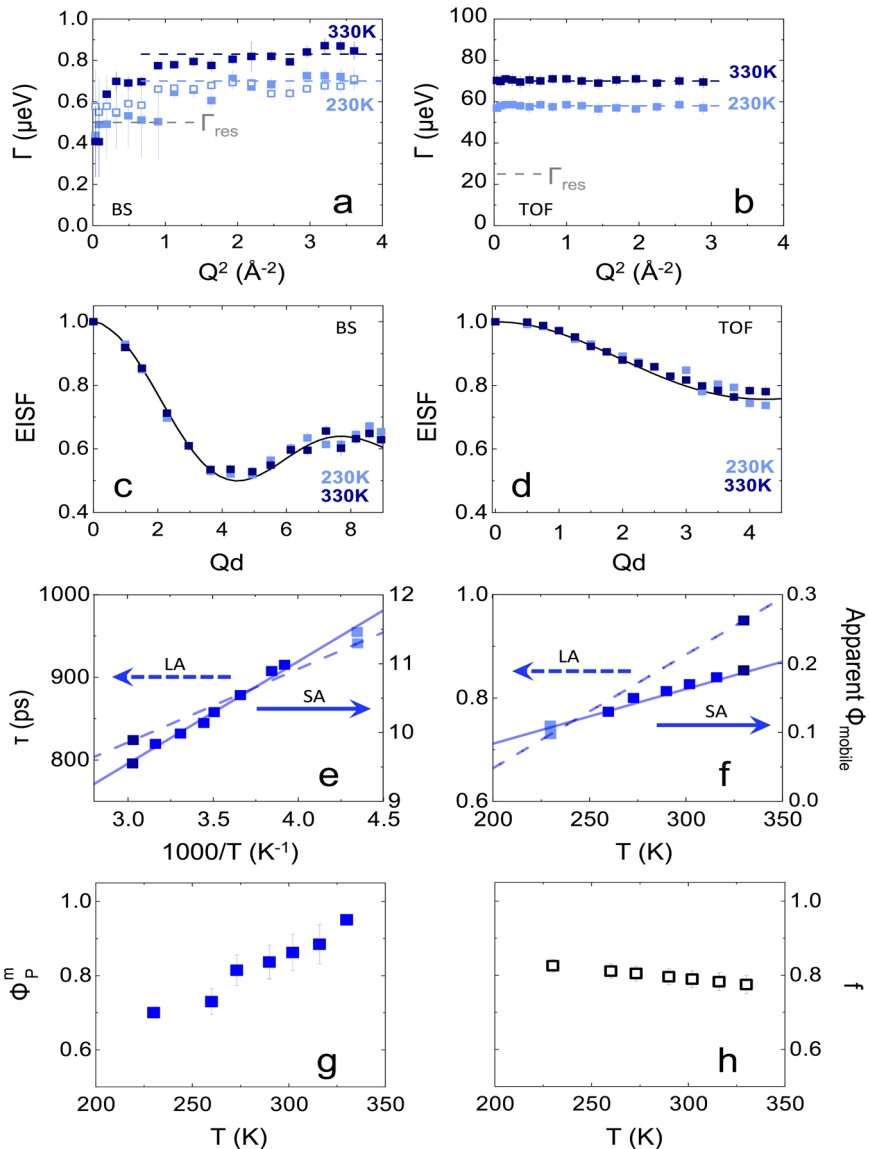

**Fig. 3 Dry PA analysis of QENS data. a, b** Linewidth HWHM (Γ) dependence on $Q^2$ measured for the dry polymer (at selected temperatures 230 and 330 K) by (**a**) BS with (half) resolution $\Gamma_{res} = 0.5\,\mu$eV, shown in gray; data acquired within $\Delta E$-range of $\pm$ 30 and $\pm$ 4.5 $\mu$eV are shown with solid and open markers, respectively; **b** $\Gamma(Q^2)$ data obtained by TOF with $\Gamma_{res} = 22.5\,\mu$eV. The error bars correspond to the fitting uncertainties of the linewidth. **c, d** EISF as a function of $Qd$ (Eq. (3)), accounting for the fraction of mobile protons in the dry PA shown in Fig. 2, obtained from BS and TOF data, respectively. The model fit for an out-of-plane, two-state, flip of the MPD ring computed from Eq. (2) (detailed in Supplementary Note 4: Model 1) is shown as a black solid line. **e, f** The corresponding rotational relaxation times computed by $\tau_0 = \hbar/\Gamma_P$, and fraction of mobile protons (indicated in Fig. 2c), as a function of temperature, observed within BS (~ns) and TOF (~10 ps) timescales. **g** Mobile proton fraction $\phi_P^m$; **h** Fraction undergoing both *SA* and *LA* motion (*f*). The error bars correspond to the maximum range compatible with the data and model employed. Source data files provided.

water, which is well described by a jump-diffusion model[29,37]:

$$S_W^{tr}(Q, \omega) = \frac{1}{\pi} \frac{\Gamma_W^{tr}}{\omega^2 + \left(\Gamma_W^{tr}\right)^2} \qquad (7)$$

where the linewidth (half-width at half-maximum) is

$$\Gamma_W^{tr} = \frac{D_{tr} Q^2}{1 + D_{tr} Q^2 \tau_0} \qquad (8)$$

and $D_{tr}$ is the diffusion coefficient and $\tau_0$ the residence time between effectively "instantaneous" jumps, whose mean jump length can be estimated by $l = \sqrt{6 D_{tr} \tau_0}$. In the long-distance limit (Q $\rightarrow$ 0), or at sufficiently high temperatures, the expected Fickian diffusion result $\Gamma_W^{tr} = D_{tr} Q^2$ is recovered, as expected. At low temperatures, between 150 and 230 K, the contribution of

water to the experimental signal becomes vanishingly small and data can be modeled considering only the polymer segmental relaxation. At higher temperatures ($\geq$240 K), the additional quasi-elastic broadening, due to water translations, is observed and eventually dominates the signal. The dependence of $\Gamma_W^{tr}$ follows the jump-diffusion model given by Eq. (8); however, a constant Γ plateau emerges at low-Q for the lowest temperatures (<290 K) shown in Fig. 4d. This is a signature of local confinement and further refinement of the model is needed, detailed next.

**Spatially confined water.** Confined water generally exhibits longer residence times for translational diffusion, deviations in bulk structure[40,41], and additional diffusive processes[42,43]. Dynamic confinement results in the appearance of an elastic component in

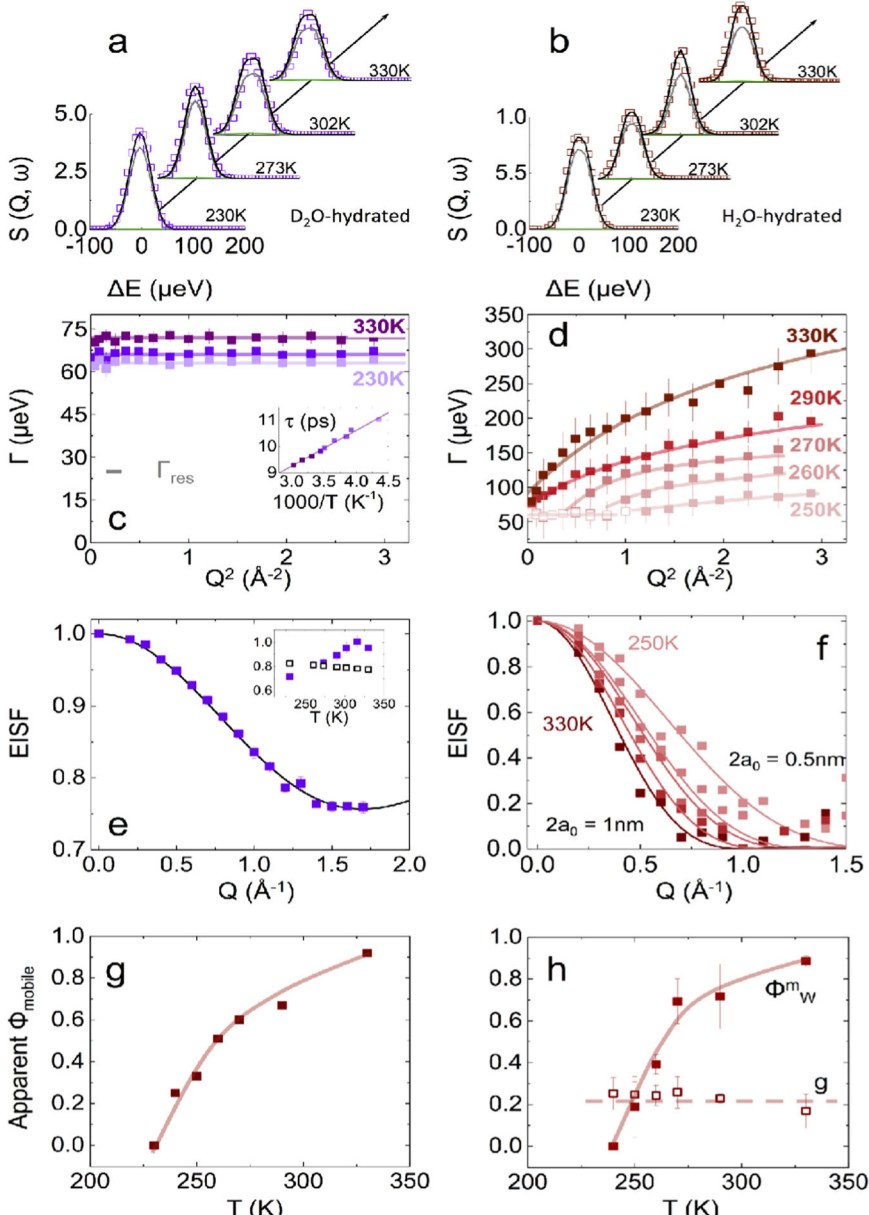

**Fig. 4 Hydrated (vapor) PA analysis of QENS data.** $S(Q, \omega)$ data and model fit, measured by TOF, at 45 $\mu$eV resolution, for the PA hydrated polymer in (**a**) in $D_2O$ and (**b**) $H_2O$ vapor, at selected $Q = 1.5$ Å$^{-1}$ and temperatures indicated. **c** HWHM for PA hydrated polymer relaxation (and associated times in inset) measured in $D_2O$, at selected temperatures and (**d**) of the water translational dynamics, measured from the second Lorentzian profile; the gray horizontal line shows the (half) instrumental resolution ($\Gamma_{res} = 22.5$ $\mu$eV). The error bars correspond to the fitting uncertainties of the linewidth. **e** EISF accounting for the mobile protons fraction at 330 K, for the out-of-plane, two-state flip depicted in Fig. 2c–e (detailed in Supplementary Note 5: Model 2), showing the fraction of mobile protons in the inset. **f** EISF for the diffusion of water molecule inside of a sphere or radius $a$, computed from the water signal in the PA hydrated by $H_2O$ vapor (second Lorentzian) and Eq. (8) (detailed in Supplementary Note 6: Model 3). **g** Overall, apparent proton mobile fraction computed from the TOF signal of hydrated PA. **h** Fraction $\phi_W^m$ of water molecules undergoing $tr$ diffusion, and $g$ fraction of those which undergo both $lr$ and $tr$ diffusion. The error bars correspond to the maximum range compatible with the data and model employed. Source data are available at https://doi.org/10.5291/ILL-DATA.9-11-1809 and https://doi.org/10.5291/ILL-DATA.9-11-5311718, and provided as a Source Data file.

the dynamic structure factor[37,44], and in the effective invariance of the linewidth $\Gamma$ at low-$Q$. We thus re-write Eq. (7) as:

$$S_W^{tr}(Q, \omega) = A_{0,W}^0(Q)\delta(\omega) + (1 - A_{0,W}^0(Q))\frac{1}{\pi}\frac{\Gamma_W^{tr}}{\omega^2 + (\Gamma_W^{tr})^2} \quad (9)$$

where, for the diffusion of a point particle inside a sphere of radius $a$:

$$A_{0,W}^0(Qa) = \left(\frac{3j_1(Qa)}{Qa}\right)^2 \quad (10)$$

where $j_1$ is a spherical Bessel function. The initial $\Gamma_w^{tr}(Q \to 0) \equiv \Gamma_0 = 4.33D_{loc}/a^2$, up to $Q \simeq \pi/a$ beyond which it follows Eq. (8), as the intensity of the elastic peak vanishes. $D_{loc}$ defines a local diffusion coefficient within the confinement region (approximated to a sphere) which, according to the chemistry of the surface and hydration level, can be either higher or lower than the macroscopic diffusion coefficient $D_{tr}$[40,45]. The confinement radius $a$, on the other hand, can be estimated from both the experimentally measured EISF, and from the maximum $Q$-value beyond which $\Gamma$ is no longer constant. While Dianoux et al.[46] extended the model to

anisotropic confinement, including a cylinder, we found the simpler spherical model to suffice in describing all data within measurement uncertainty.

The EISF for water translational diffusion as a function of temperature is shown in Fig. 4f, and a model fit to Eq. (10) yields a confinement diameter ($\equiv 2a$) ranging from 0.5 to 1 nm, from 250 to 330 K. To our knowledge, this is the first direct experimental evidence for *dynamic* confinement of water in PA membranes. This effective pore size is consistent with the length scale obtained from the maximum $Q$-value beyond which $\Gamma$ is no longer constant (Fig. 4d). Further, these dimensions are commensurate with the characteristic length scale of the rotational relaxation of the polymer (Fig. 2c–e), although a possible coupling between polymer relaxation and water dynamics can not be demonstrated based on the present results. These dimensions are also compatible with experimental (PALS and SAXS) and modeling reports of NF/RO membranes, which suggest pore radii in the range $\simeq 2.1$ to $\simeq 5.2$ Å[21,22,32], as well as the effective sizes implicit in membrane rejection data for desalination.

Analyzing the linewidth $\Gamma(Q)$ in Fig. 4d, corresponding to $S_W^{tr}(Q, \omega)$, well-resolved by TOF, with Eq. (10) yields the confined water translational diffusion coefficient ($D_{tr}$) as a function of temperature, which is found to vary from $\simeq 0.5$ to $\simeq 2.4 \times 10^{-5}$ cm$^2$ s$^{-1}$ between 250 and 330 K, respectively (Supplementary Table 1).

Employing high-energy-resolution BS, we resolve principally $S_W^{lr}(Q, \omega)$ while the term $S_W^{tr}(Q, \omega)$ appears now effectively as a flat background. Analysis of BS data fixing the $S_W^{tr}(Q, \omega)$ contribution obtained from TOF (extremely broad in this energy window) or, alternatively, a flat $Q$-dependent background $B(Q)$ yields similar results within measurement uncertainty. The temperature-dependent profiles $S_W^{lr}(Q, \omega)$ and fitted $\Gamma_W^{lr}$, provided in Supplementary Fig. 12 are similarly well described by Eqs. (7) and (8), from which a long-range translational diffusion ($D_{lr}$) coefficient for water wihin the PA matrix can be estimated. Values of $D_{lr}$ are found to be approximately one order of magnitude smaller than $D_{tr}$ at the same temperature. Further, from the initial plateau of $\Gamma_W^{tr}$ obtained by TOF, shown in Fig. 4d, a third (faster) local diffusion coefficient, within the confinement volume, can be estimated, $D_{loc}$, and is found to increase with temperature (from 1.9 to $3.4 \times 10^{-5}$ cm$^2$ s$^{-1}$ between 250 and 270 K).

These water-diffusion mechanisms are tentatively illustrated in Fig. 5a, based on the picture of water diffusion through a spatial heterogeneous medium of "network" and "aggregate" pores, and the corresponding coefficients plotted in Fig. 5b, yielding a "diffusion map" for water confined within PA membranes, which we experimentally resolve for the first time. For comparison, we have also included the values for the self-diffusion of bulk water extended into the supercooled regime[29,30,47], as well as those computed by MD simulations of water in PA membranes[11–13,22,25]. Since water diffusion in PA confinement remains Arrhenius in the entire temperature range, while bulk water exhibits non-Arrhenius diffusion toward the supercooled regime, water diffusion is slowed down within PA membranes above 0 °C (2.4 vs. $4.8 \times 10^{-5}$ cm$^2$ s$^{-1}$, at 330 K) but is actually *faster* than bulk water below 0 °C, where the two cross-over, as shown in Fig. 5b (Supplementary Table 1). Further, we find that the diffusion jump length $l$ (Fig. 5c) and residence time $\tau_0$ (Fig. 5d), associated with translations $tr$ remains Arrhenius down to supercooled (sub-zero) temperatures, differing markedly from bulk water behavior. The mobile water fraction $\phi_W^m$ (Eq. (6) and Supplementary Notes 4–6) decreases with temperature, until it is no longer detectable, as indicated by the red-shaded region.

This behavior is reminiscent of that of confined water in hydrophilic spinodal Vycor glass of ~5 nm pore size, under full hydration conditions[40,48].

Analysis of high-resolution BS data, at selected temperatures, provides an estimate for the small fraction ($\phi_W^m g \sim 15\%$) of water molecules undergoing diffusing over longer timescales and correlation lengths, with $D_{lr} \simeq 0.23 \times 10^{-5}$ cm$^2$ s$^{-1}$ at 330 K, more than one order of magnitude lower than bulk water diffusion (Fig. 5b). Our decoupled method of analysis of the various diffusive motions (termed localized, translational and long range), is made possible by the separation of timescales corresponding to $tr$ and $lr$ motions which can be well-resolved by the complementary TOF and BS energy windows, using a single model for analysis (Eq. (5)); further, it benefits from the polymer dynamics previously resolved by the PA-D$_2$O measurement series under the same conditions.

We have associated with the various diffusion processes to the spatially heterogeneous nature of PA membranes, characterized by aggregate and network pores, and supported by MD simulations, thus corresponding to a space-dependent distribution of diffusion coefficients (reflected by the $Q$-dependence of the processes identified). The spread of diffusion coefficients observed by MD likely reflects sub-diffuse behavior of water within this confined ("crowded") system, and other frameworks for analysis can potentially be compatible with the data. Nevertheless, our work reports the first experimental evidence of the distribution of water-diffusion processes within PA membranes relevant for RO water treatment, providing a comprehensive relaxation map of water dynamics over pico to tens of nanoseconds, and enabling the validation of transport simulations at the molecular scale, as well as reaction models for PA membrane formation by IP.

**Resolving controversy with previous QENS measurements**. Previously reported $D_{tr}$ QENS data[33–35] are also shown in Fig. 5b and, surprisingly, these largely overlap with those of bulk water, in particular at higher temperatures. We hypothesize that this puzzling result might be caused by the membrane hydration procedure employed, viz., direct immersion in liquid water and careful pad drying. The presence of corrugations, "lagoons" or comparatively large voids[49–51] might be expected to inadvertently lead to a considerable excess of bulk-like water. We, therefore, repeated our TOF experiment series on water-immersed PA membranes (Supplementary Fig. 15), and plot the obtained $\Gamma_{tr}$ in Fig. 5e. Gravimetrically, we find a considerably larger water fraction ($\simeq 50\%$ w/w, or $\lambda \simeq 19$) compared to that found in vapor hydrated membranes ($\simeq 14\%$w/w, or $\lambda \simeq 3$). The proton fraction of water ($\varphi_W$) thus increases to 0.76 (from 0.33 as in the case of PA vapor-hydrated (Supplementary Note 7). Under these conditions, we also recover the result of water-diffusion coefficients analogous to those of pure bulk water[29,30]. We, therefore, confirm that the apparent insensitivity of water dynamics within PA membrane confinement found in previous work derives from a sample preparation protocol that results in a considerable bulk water fraction, thus preventing the direct measurement of confined water that can be accessed with the vapor-hydration procedure.

In immersed membranes, a weighted average of "bulk" and confined water populations is thus probed and, given the large fraction of the former, this value dominates, while small deviations are observed below freezing, as bulk water starts to crystallize (Supplementary Fig. 15). A comparison of calorimetric data (Fig. 5f) of the vapor hydrated and liquid immersed PA membranes corroborates our assertion: while the immersed membrane (light blue) exhibits two relatively broad crystallization

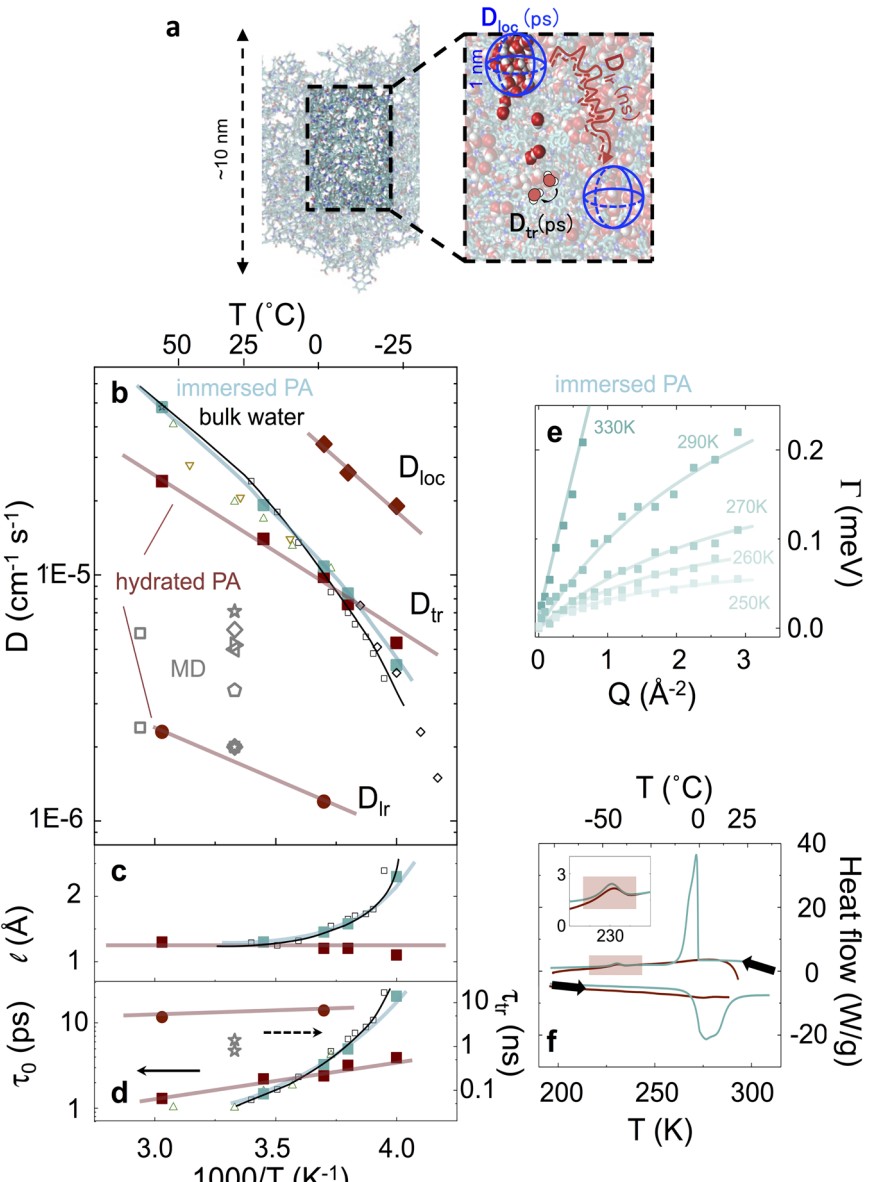

**Fig. 5 Confined water-diffusion map and contrast with bulk water. a** Schematic of the experimentally resolved diffusive modes of water confined within PA membranes, namely localized diffusion (blue sphere), translational diffusion, and long-range diffusion (tortuous red arrow). **b** Translational water-diffusion map showing $D_{tr}$, $D_{lr}$ and $D_{loc}$ as well as "bulk" water diffusion. Present data: water dynamics within PA membranes prepared by vapor hydration (■ TOF and ● BS), and immersed in liquid $H_2O$ (■ TOF); previous QENS measurements of water in PA membranes (△[34]; ▽[35]), as well as bulk water (□ TOF[29], ◇ pulsed-laser-heating[47], ★ NMR[30]). Diffusion coefficients obtained by MD simulations (★[25], ○[10], ◁[11], ◇[22,26], ▷[12], □[28]) are also included for comparison. **c** Jump-diffusion length $l$ and (**d**) residence time $\tau_O$ (labeled as above). **e** variation of the fraction of mobile protons in PA hydrated by $H_2O$ vapor as a function of temperature. **f** Lorentzian width ($\Gamma$) obtained for water in liquid $H_2O$ immersed PA, exhibiting analogous "bulk" behavior in contrast to vapor hydrated PA membranes (Fig. 4d). **g** DSC trace of vapor hydrated (red) and liquid immersed (green) PA membranes showing, upon cooling, the crystallization of water near bulk temperatures for the latter. Source data files provided.

peaks, around −45 °C and 0 °C, the vapor hydrated membrane (dark red) shows only the lower temperature transition, which we associate with confined water, coinciding with the shaded temperature range in Fig. 5f. The pronounced "bulk" water crystallization (and broad melting) feature observed in liquid immersed PA membranes is consistent our QENS findings.

**Benchmarking with MD simulations.** MD simulations generally reveal bimodal diffusion processes and timescales, associated with the heterogeneous PA membrane structure, distance from the membrane-bulk interface, and related mechanisms for water

diffusion[22,25,27,28]. These are discussed in terms of water at the interface or within the interior of the membrane, or water within network and aggregate pores[11], constituting effectively a trimodal profile, including bulk water reservoirs and "lagoons". The slowest, long-range diffusion coefficient $D_{lr}$ reported experimentally in our study is in good agreement with that of confined water from the majority of MD simulations[22,25,28], while the order of magnitude of the usual translational diffusion $D_{tr}$ is captured but not its exact value. Further, the "local" diffusion coefficient $D_{loc}$, defined within the confining volume, is generally not computed from MD. Our experimental diffusion map provides direct quantification of the multimodal dynamic spectrum

of confined water within PA RO membranes which will enable henceforth the validation of simulated transport mechanisms and timescales.

**Consequences for macroscopic membrane flux**. We finally consider how our diffusion data can be predictively related to the macroscopic performance observed in PA RO membranes, and specifically on the water permeance (L m$^{-2}$ h$^{-1}$ bar$^{-1}$, defined by the flux normalized by the applied hydraulic pressure. Estimating transport across a PA membrane, with active and inactive, connected and disjoined, permeation pathways, is evidently challenging, and work is still needed to resolve membrane structure at the nanoscale[17,18,52] at in or near *operando* conditions. A number of engineering models[9] take into account membrane observables, including water uptake, an "effective" thickness, and mass or density changes (from which a "void fraction" and water partition coefficient are computed) to infer molecular diffusion parameters from permeance data, subject to a series of assumptions[49]. While gravimetric approaches (namely quartz crystal microbalance, QCM) and transmission electron microscopy (TEM) can be employed to approximate uptake and effective thickness in corrugated membranes, we opt to employ planar nanofilm membranes[23,24] for which film thicknesses can be accurately determined from neutron reflection or atomic force microscopy, and perform 'inverse' calculations. Specifically, we estimate macroscopic permeance of representative PA membranes from molecular diffusion data and utilize the *same* diffusion coefficients, obtained in this work, to calculate the macroscopic performance of *diverse* membranes of similar chemical structure. We consider membranes with different void fractions and water uptake and take a weighted average of $D_{tr}$ for diffusion through the "bulk" polymer and $D_{lr}$ for "pore"-diffusion, to obtain a *forward* prediction for water permeance (Supplementary Note 8 and Supplementary Fig. 17), that agrees well with PA macroscopic data fabricated under analogous conditions. For a range of illustrative membranes of varying TMC:MPD reactant stoichiometry, Supplementary Fig. 17 shows the measured and calculated permeances with void fractions ranging from 7 to 16% obtained from QCM and AFM (for dry and hydrated mass, and effective thickness). For the membrane stoichiometry investigated here, we compute a permeance of 1.4 LMHBar, which compares favorably with 1.20 ± 0.05 LMHBar measured experimentally. Despite the simplicity of the model, the agreement between the estimates appears consistent with our decoupled model for analysis of the neutron data.

## Conclusion

Extensive QENS/INS measurements at complementary resolutions, on dry and hydrated polyamide membranes, employing H/D labeling of water and contrasting hydration procedures, provide unprecedented insight into the dynamics of the polymeric network and the confined water, required for an evidence-based molecular understanding of transport in RO membranes. The polymer network is found to exhibit rotational relaxations that are well described by a partial flip of the MPD ring with two distinct amplitudes (15° and 40°), in agreement with MD simulation. We decouple the local polymer relaxation from the water translational diffusion, which we find to be multimodal and elucidate the transport mechanisms and associated diffusion coefficients characterizing water confined within the PA matrix. To access the water dynamics in confinement, we hydrate the membranes via the vapor phase, building on our previous synthetic work[24] and examination of hydration kinetics[23].

We find that water translational diffusion is well described by a jump-diffusion model with a diffusion coefficient $D_{tr}$ which is

approximately half of that of bulk water, crossing over at lower temperatures, due to the fact that PA-confined water diffusion remains Arrhenius down to temperatures where supercooled bulk water diffusion becomes strongly non-Arrhenius. The jump-diffusion length remains $l \simeq 1.2$ Å in agreement with bulk water near ambient conditions, while residence time $\tau_0$ increases upon confinement yet exhibits Arrhenius behavior at all temperatures, by contrast with supercooled water. We obtain a direct evidence of confined translational diffusion at an effective length scale of $\simeq 1$ nm, compatible with the inferred "pore size" from separation measurements. Further, this dimension is comparable to the amplitude of polymer rotational relaxation of MPD rings, which might indicate a coupling between polymer relaxation and water dynamics within PA membranes. We obtain a comprehensive water-diffusion map, spanning two orders of magnitude in $D$ and approximately 100 °C in temperature, encompassing the operating conditions of RO membranes, and experimentally resolving bulk, local, confined, and long-range diffusion processes. We expect that our findings will enable the development and validation of accurate molecular models for high-performance RO membranes and the predictive computation of their macroscopic, engineering performance.

## Methods

**Materials**. Trimesoyl chloride (TMC) 98% and m-phenylenediamine (MPD) flakes ≥99% were purchased from Sigma-Aldrich Ltd. (Gillingham, UK). N-hexane ≥98.5% was supplied by VWR International Ltd.; DI water at 18 MΩ residual-specific resistance was employed. D$_2$O (>99.7% D) was supplied by Goss Scientific Instruments Ltd. (Cheshire, UK).

**Membrane fabrication and hydration**. Fully aromatic PA membranes were fabricated as a crumpled film via interfacial polymerization (IP) following procedures reported previously[23,24] at the liquid interface. In a 1-L reservoir containing the aqueous phase (250 mL, 10wt% MPD), the organic solution (250 mL of n-hexane with 0.5wt% TMC) was carefully poured laterally to form a planar, stable interface. The reaction was allowed to take place over 1 min, after which the crumpled polymer film was collected from the interface, and thoroughly washed with n-hexane to remove any unreacted TMC. To ensure the complete removal of any water and/or solvent residues, the membranes were dried under vacuum for 18 h at 30 °C. Films were then cold-pressed into coupons of 4 × 5 cm and 300–400 μm thick. Membrane stoichiometry was estimated from XPS measurements[24,53]). We opted to employ crumpled films instead of stacked nanofilms in order to attain QENS statistics required to quantify the multimodal diffusion coefficients (within reasonable timescales), and minimize interstitial (bulk) water between films, which could invalidate our approach; finally, we believe that this procedure approximates more closely the ubiquitous RO membranes fabricated by IP. Three different hydration procedures were compared: (i) fully dried, measured immediately after removal from a vacuum; (ii) hydrated in water vapor, H$_2$O or D$_2$O, (using in a humidity chamber where the hydration level is controlled by the temperature difference between the liquid water reservoir and the sample chamber set to room temperature[23]); and (iii) hydrated by immersion in liquid water. Following 18 h of exposure, the PA membrane reached full hydration, as measured gravimetrically (detailed in SI, section 4). Procedure (iii) was carried out as a control and comparison with previous work[33,54]: after the procedure (ii), the membrane was dipped in liquid water (either H$_2$O or D$_2$O) and allowed to soak for a further 4 h; the excess surface water was then gently pad dried to attempt its complete removal. Membrane samples were weighted before and after each step to quantify water uptake (detailed in SI). The water uptake is estimated gravimetrically to be 14–15% w/w. For procedures (i), (ii), and (iii) we obtain water/polymer molar ratio of approximately $\lambda = 0$, 3, and 19, respectively.

**Quasi-elastic neutron scattering**. QENS experiments were performed using both a TOF (IN5) and a high-resolution BS (IN16B) spectrometer at the Institut Laue Langevin (Grenoble, France). BS experiments were performed using the standard unpolished strained Si(111) monochromator and analyzer corresponding to an incident wavelength of $\lambda = 6.271$ Å (covering a $Q$-range between 0.2 and 1.8 Å$^{-1}$) and an instrumental energy resolution of 1 μeV (full-width-half-maximum obtained by measuring the dry sample at 2 K): this configuration allows probing motions in the ns timescale ($0.4 < \tau < 1$ ns). Dynamic processes with longer characteristic times appear "elastic" while faster processes contribute to a background in the data. Elastic and inelastic fixed window scans (EFWS and IFWS)[38] were recorded at $\Delta E = 0$ and 2 μeV energy at a heating rate of 0.13 K min$^{-1}$ from 2 to 380 K, and QENS profiles were acquired at selected temperatures. In order to investigate faster processes (which would appear as background in a BS

experiment), complementary TOF experiments were performed from 150 to 330 K using $\lambda = 6$ Å ($0.2 \leq Q \leq 1.6$ Å$^{-1}$) and energy resolution of 45 $\mu$eV, thus probing motions in the ps timescale ($1 < \tau < 50$ ps). Samples were contained within a flat aluminum can ($4 \times 5$ cm; 0.6 mm inner thickness) to achieve a neutron transmission of 90% or higher, thus minimizing multiple scattering. Vanadium reference and empty can measurements were carried out for data normalization and sample correction, respectively.

## Data availability

Neutron TOF and BS data underlying Figs. 2a, b, 3a, b, 4a, b, and Supplementary Figs. 4, 5, 7, 8a–c, 10, 12, 13a–d, i–l, and 15 are available at https://doi.org/10.5291/ILL-DATA.9-11-1809 and https://doi.org/10.5291/ILL-DATA.9-11-1718. Source data are provided with this paper.

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

## Acknowledgements

We thank the BP International Centre for Advanced Materials (BP-ICAM) for financial support, and J. Muscatello, E. Muller, and A. Sutton for useful discussions. J.T.C. thanks the Royal Academy of Engineering (UK) for a research chair, and EPSRC (EP/L020564/1) for funding. We also thank the Institut Laue Langevin (Grenoble, France) for neutron beamtime and the Partnership for Soft Condensed Matter (PSCM) for access to support laboratories.

## Author contributions

J.T.C. and F.F. designed research; F.F., B.F., and M.N. performed measurements and F.F. and J.T.C. analyzed and interpreted data, assisted by B.F. and A.G.L.; J.T.C. and F.F. wrote the paper.

## Competing interests

The authors declare no competing interests.
