## [Peer Review File · Nature Communications]

nature portfolio

Peer Review File

Draft OnlyEditorial Note: This manuscript has been previously reviewed at another journal that is not operating a transparent peer review scheme. This document only contains reviewer comments and rebuttal letters for versions considered at *Nature Communications*.

REVIEWER COMMENTS

Reviewer #1 (Remarks to the Author):

I reviewed this paper before and still have the same reservations regarding its suitability for a general science venue.

Reviewer #3 (Remarks to the Author):

The manuscript provides some new and interesting results related to the confined water dynamics in polyamide RO membranes. The work is supported by some of the latest advanced membrane characterization studies and may facilitate deep understanding of the transport mechanisms involved in RO membranes. The authors have properly addressed all my review comments (provided through Nature Nanotechnology). Therefore, I recommend the publication of the manuscript.

Review of Foglia et al, submitted to Nature Com.

Several points raised by the reviewer were answered by the authors who provided useful and convincing new information. Some figures were also re-elaborated to be clearer, which is also appreciated. However, the reviewer believes that the data analysis is still lacking some precise descriptions and, therefore, the conclusions are not fully supported in a comprehensive manner. Only by providing the requested details and answers can the work be publishable on Nat. Com.

The analysis relies on the detailed analysis of two sets of data, namely BS (backscattering, IN16B) and Time-of-Flight (TOF, IN5). The resolutions are very distinct, therefore “slow” motions are probed by BS while fast motions are probed on IN5. Nevertheless, the “slow” motions are present in the elastic line on IN5, while the fast motions are present in the background of the BS measurements. The authors decided to perform a decoupled analysis, which means that they do not have one single analytical function that they fit to the whole set of data, but they assume that they have a double-component dynamics (Eq 3 ; Eq 13 of the SI): one term accounting for polymer dynamics (Lorentzian 1), and one term accounting for water dynamics (Lorentzian 2). The respective amount of these terms is weighted by using the volume fraction of polymer and water. For each species, then, a fraction of mobile and immobile protons is introduced to balance the elastic and quasielastic contributions. This approach can be acceptable, although, of course, it is a simplified frame that raises some questions, in particular relative to the claim of a multimodal dynamics, which is needed to be clarified by stating the underlying hypothesis.

My main concerns are three-fold (detailed comments are provided below):

- **Fig. 5 is misleading.** The nature of proton motions and corresponding populations must be better introduced and explained in the text, and sketched differently.
- **The nature of slow motions is not unambiguously supported by the data.** The text introducing this analysis is confusing, fitting parameters are missing, the reviewer asks some clarifications.
- **Multi-modal dynamics is not clear.** What does it mean ? Does it refer to polymer vs water ? Does it refer to interfacial vs bulk ? Does it refer to slow vs fast dynamics ? Does it refer to motions in pores or through polymer ? The various possibilities must be defined by the authors as a preamble of their paper, in a (missing) paragraph where they explain their assumptions and methodologies, and it must be highlighted again in the conclusions.

Detailed comments to be addressed:

PRINCIPLE OF THE QENS ANALYSIS – QUESTIONS ON MULTI-MODAL DYNAMICS

A - In the dry and D2O-hydrated samples, the authors show that the polymer dynamics is the only constituent they need to reproduce the data. The reviewer is now convinced by this result. Hence, the polymer component is determined in the H2O-hydrated data and used as a fixed input to better evaluate the water dynamics.

B - Most of the H2O-hydrated results rely on the analysis of the TOF data. The HWHM dependence of the water dynamics Lorentzian (component 2) is analysed in the frame of both the confinement in a sphere model, yielding the confined diffusion coefficient D_{loc} (analysis of low-Q part of the HWHM profile, i.e. a plateau obtained at low temperatures) and the jump diffusion model (deviation from Fick's law at high Q, that allows to determine “another” diffusion coefficient called D_{tr} , as well as

typical time τ_0 and length l , the latter being completely correlated to the two previous ones). This analysis, as raised in the first reviewer's report, is not optimized (using two models to describe the *same* HWHM dependence obtained by a single-Lorentzian fit is an approximation) but can be acceptable, keeping in mind that it provides information of **the same protons/water molecules** "seen" at different scales (low and high-Q parts of the profile). Therefore, D_{loc} and D_{tr} **MUST** be associated to the same type of motions and same type of translational diffusion. At this point, therefore, the scheme shown on Figure 5 is not consistent. The reviewer suggests to modify it and make clear that D_{loc} corresponds to the fast diffusion of a proton inside a confining sphere (as drawn), while D_{tr} corresponds to the jump diffusion of the same proton. The concentric circles added by the authors on Fig.5 are not understandable based on their assumptions. The "proton jumps" correspond to the elementary step of the long-range diffusion mechanism that is evidenced here (because data are treated in the frame of a jump diffusion model), and this elementary process is ALSO THE ELEMENTARY STEP of the confined motions inside the sphere, by construction (one single-Lorentzian, one single process). To summarize:

- 1) decoupling the "concentric circles" motion from the "confining sphere" in Fig. 5 is misleading.
- 2) one type of motion is evidenced from the TOF data (one fraction of mobile protons, one characteristic time τ_0 , one elementary jump length l , two diffusion coefficients because the same dynamics are considered at different length-scales (low Q and high Q) : motions are fast in the 1nm sphere, but slowed down when the proton is travelling on longer distances ($D_{tr} < D_{loc}$). This is why the reviewer, in his first report, raised the point of "sub-diffusive" behavior, or, said in a different manner, a space-dependent distribution of diffusion coefficients. This point should be at least mentioned by the authors. By the way, this is somehow reflected in the final graph where the authors plotted their diffusion coefficients together with the ones from literature, in particular MD. The spread in values from MD is exactly reflecting the sub-diffusivity... The reviewer is not asking to re-do any analysis in a sub-diffusive analytical framework but to mention this aspect when they comment these values. In fact, the reviewer is convinced that multi-modal is not adapted to describe the complex behavior in these membranes and rather multi-scale, or scale-dependent, which, at least, puts all data in a bigger picture based on a physical mechanism that is well-known in situations of harsh confinement. The reviewer also suggests that the translational diffusion coefficient D_{tr} may correspond to long-range diffusion between confinement domains, e.g. rather correspond to the "red line" motions drawn on the figure, which is associated to the BS data by the authors without strong justification (see point C).

C – Once the data from TOF were analysed, the authors analyse the BS data. They, again, use a two-component function, one Lorentzian for the polymer dynamics, one for the water dynamics. The latter is completely decoupled from the TOF analysis (it is in fact Lorentzian 3). It can be accepted, if clearly stated. The need for 2 components is demonstrated in Fig. 12 of the SI and is convincing (of course, a background is needed, that would be the TOF water signal). However, the determination of the diffusive nature of the observed slow water motions is not convincing (Figure.SI-12, panels q-r). The polymer Lorentzian is an almost flat HWHM (blue and purple, q), with some small deviations at small Q that can be neglected because it comes close to the resolution and fits are difficult (what about the background variations with Q ??). However, the HWHM profiles obtained from the water component (red) are somehow very similar. The reviewer guesses that the dashed black line in (r) is the fit using a jump diffusion model (not described in the legend), from which values of the third diffusion coefficient labeled D_{lr} are extracted. This profile is resembling the one from the polymer (just shifted to higher HWHM, faster motions). The diffusive behavior is questionable. It could well be

a localized motion, with not much reliability given on the first 2-3 low Q-values results. What are the values of the mobile fraction of protons and the elementary times related to this slow jump diffusion mechanism ? What are the associated jump lengths and how do they compare to TOF values of l ? Comparison between the fraction of BS mobile water protons with TOF mobile water protons is mandatory to have insights into the amount of water involved in this second type of dynamics. The question is: are these protons the same moving much slower, or are these protons located elsewhere and not exchangeable within the timescale ?? By answering this, one can then correctly refine the scheme of Fig. 5. and seriously claim about decoupled dynamics, multi-modal dynamics, and link with MD findings. It could be that there are several paths for diffusion in the material due to the hierarchy of pores and scales, but, as it is, the authors discussions are only speculative.

Globally, the reviewer asks for a better description of the methodology (specially for non QENS-specialists, such that the findings are accessible to a broad audience) :

Authors text with modifications	Reviewers suggestions
Line 315 to 373. Analyzing the linewidth $\Delta(Q)$ in Fig. 4d (TOF) with Eq. 6 yields the confined water translational diffusion coefficient (D_{tr}) as a function of temperature, which are found to vary from 0.5 to $2.4 \times 10^{-5} \text{ cm}^2 \text{ s}^{-1}$ between 250 and 330 K, respectively (Supplementary Table 1). Employing high energy resolution BS, we identify an additional, slower, long range translational diffusion (D_{tr}) process within the matrix, differing in value by approximately one order of magnitude. Further, from the initial plateau in Fig. 4d, a third (faster) local diffusion coefficient, within the confinement volume, can be estimated, D_{loc}, and is found to increase with temperature (from 1.9 to $3.4 \times 10^{-5} \text{ cm}^2 \text{ s}^{-1}$ between 250 and 270 K). These water diffusion mechanisms are illustrated in Fig. 5a and the corresponding coefficients plotted in Fig. 5b, yielding a 'diffusion map' for water confined within PA membranes, which we experimentally resolve for the first time. For comparison, we have also included the values for the self-diffusion of bulk water extended into the supercooled regime (26, 27, 43), as well as those computed by MD simulations of water in PA membranes (11–13, 19, 22). Since water diffusion in PA confinement remains Arrhenius in the entire temperature range, while bulk water exhibits non-Arrhenius diffusion towards the supercooled regime, water diffusion is slowed down within PA membranes above $0 \text{ }^\circ\text{C}$ (2.4 vs $4.8 \times 10^{-5} \text{ cm}^2 \text{ s}^{-1}$, at 330 K) but is actually faster than bulk water below $0 \text{ }^\circ\text{C}$, where the two cross-over, as shown in Fig. 5b (Supplementary Table 1). Further, we find that the diffusion jump length l (Fig. 5c) and residence time τ_0 (Fig. 5d), remain Arrhenius down to supercooled (sub-zero) temperatures, differing markedly from bulk water behavior. The mobile water fraction f_{mw} (Eq. 4 and Supplementary text 4-6) decreases with temperature (Fig. 5e), until it is no longer detectable, indicated by the red shaded region. This behavior is reminiscent of that of confined water in hydrophilic spinodal Vycor	At this stage, mentioning the BS analysis is confusing. Proceed with the TOF description first, and then move to BS. Suppress here "employing...magnitude". At this stage, D_{loc} is the "second 'diffusion coefficient, and it should be precised that it corresponds to the SAME PROTONS as the ones showing long-range translational diffusion." Keep this for later (after BS results being described) : "yielding a 'diffusion map' for water confined within PA membranes, which we experimentally resolve for the first time."

glass of approximately 5 nm pore size, under full hydration conditions (36, 44).
 In order to extend our measurements to longer timescales, up to 10s of ns range, and probe long-range diffusion, QENS data of H₂O-hydrated PA were acquired in high-resolution BS at selected temperatures. This analysis reveals a population of confined water diffusing over longer timescales and correlation lengths, with D_{lr} 0.23 10⁻⁵ cm²s⁻¹ at 330 K, more than one order of magnitude lower than bulk water diffusion (Fig. 5b). In the above, we have adopted a decoupled analysis of the various diffusive motions (termed localized, translational and long-range), which we associate with the spatially heterogeneous nature of PA membranes (characterized by aggregate and network pores), and conceptually supported by MD simulations.

While the TOF and BS energy windows cover complementary timescales, our results remain unchanged within experimental uncertainty when we assumed a broad Lorentzian in BS (instead of a background term) or additional Lorentzians (including a possible distribution) in the TOF analysis.

Our work thus reports the first experimental evidence of the multimodal distribution of water diffusion within PA membranes relevant for RO processes, providing a comprehensive relaxation map of water dynamics over ps-10s ns, and enabling the rigorous validation of transport simulations at the molecular scale, as well reaction models for PA membrane formation by IP.

Here, it is fine to start with the BS results.

Long-range diffusion was already “probed” by TOF because Jump Diffusion is used !!!!! BS is probing slow motions.

Quantify the population ! Compare mobile fractions to TOF. Same protons or different ??? Timescales and correlation lengths are not given, please provide (in supp info).

The text in red is not complete and not added in the right place.

Section on “decoupled” analysis must be put at the very beginning of the results section.

These comments are useless. This is not the point. The true analytical function should be : $\Phi_i P * SP (Lor1) + \Phi_{iw1} * Sw1 (TOF, Lor2) + \Phi_{iw2} * Sw2 (BS, Lor3)$ (two populations of water) — Lor3 is neglected in the TOF treatment, while L2 corresponds to B(Q) in the BS data (but not correlated, not imposed to the value of integral found on TOF – whatever is used, a flat or very large broad Lor does not matter here)

Or $\Phi_i P * SP + \Phi_{iw} * (Sw1 \text{ convoluted to } Sw2)$ (one population).

The results MUST be changed by changing the analytical function because THE PARADIGM is changed. The authors need to be fair and explain that they do a first-order analysis which IMPOSES CONCEPTUALLY their main conclusions...By decoupling the BS and TOF treatment, they INTRODUCE THE IDEA OF TWO POPULATIONS AND DO NOT DEMONSTRATE IT. The reviewer is fine with it, as long as it is not hidden in the paper and announced since the beginning. May be writing the total function with 3 components in the Supp Info and explaining the hypothesis to neglect 2 or 3 alternatively in the fits.

Define multi-modal. Or use multi-scale.

Introduce here the modified scheme of Fig. 5. and sentence : These water diffusion mechanisms are illustrated in Fig. 5a, yielding a ‘diffusion map’ for water confined within PA membranes, which we experimentally resolve for the first time.”

	Rigorous ? No. Everything here is model-dependent.
Line 417-435 MD simulations generally reveal bimodal diffusion processes and timescales, associated with the heterogeneous PA membrane structure, distance from the membrane-bulk interface, and related mechanisms for water diffusion (19, 22, 24, 25). These are discussed in terms of water at the interface or within the interior of the membrane, or water within network and aggregate pores (11), constituting effectively a trimodal profile, including bulk water reservoirs and 'lagoons'. The slowest, long-range, diffusion coefficient D_{lr} reported experimentally in our study is in good agreement with that of confined water from the majority of MD simulations (19, 22, 25), while the order of magnitude of the usual translational diffusion D_{tr} is captured but not its exact value. Further, the 'local' diffusion coefficient D_{loc}, defined within the confining volume, is generally not computed from MD. Our experimental diffusion map provides direct quantification of the multimodal dynamic spectrum of confined water which will enable henceforth the validation of simulated transport mechanisms and timescales. Line 462-465 We consider membranes with different void fractions and water uptake and take a weighted average of D_{tr} for diffusion through the 'bulk' polymer and D_{tr} for 'pore'-diffusion, to obtain a forward prediction for water permeance Line 486-488 We decouple the local polymer relaxation from the water translational diffusion, which we find to be multi-modal	This paragraph is interesting although not really informative on the link the authors can make between their data and the MD findings. Do they believe that D_{lr} correspond to interfacial water (very slow) ?? What is the definition of D_{tr} in the corresponding papers cited by the authors (interfacial ? pore network ????) Do they suppose that D_{tr} correspond to computed nanoscale diffusion coeff as obtained from MD (not interfacial) ? Why is D_{loc} not computed, although it better corresponds to the usual length and timescales probed by MD, rather than the other 2 ??? To strengthen this paragraph, the reviewer asks that more details are given on the various possibilities of correlating D_{loc}, D_{tr} and D_{lr} to interfacial vs bulk, and to the pore network. Here, the authors express their assumption on the meaning of the diffusion coefficients they found, corresponding to two populations of water. See previous comments on the link with MD. Why should pore-diffusion be much lower than diffusion through the polymer ? It should rather be the opposite. On what is this assumption based ? Somehow, this hypothesis of attribution of D_{tr} and D_{lr} should be reflected on the scheme of Fig. 5 to be consistent. Define multi-modal in terms of the populations and their locations in the structure

Few additional comments :

line 12 :-a generally

line 372 : as well as

Supporting info :

Figure S2. Not clear at all. Polymer contribution is not indicated, while it is surely one of the Lorentzian components. Q value is not given. Local diffusion (D_{loc}) and translational diffusion (D_{tr}) is the same Lorentzian, therefore not well indicated by the arrows. So-called long-range diffusion (D_{lr}) should correspond to the slow dynamics ??

Figure S8. a-c : not $S(q,w)$ but $S(w)$ (integrated over Q). Correct typo in legend : ESIF (EISF)

Figure S9. for both the dry...

p14 : remain remain

Figure S10. Specify what is the component (polymer) and measurement (IN16B).

Figure S12. Specify what is the dashed black line in (r) and refer to the equation used and parameters extracted from it.

Figure S13. Specify the type of measurement (TOF) from which data in a-f are extracted.

Draft Only

Reply to reviewers:

“Multimodal confined water dynamics in reverse osmosis polyamide membranes” (NCOMMS-20-01654B)

Fabrizia Foglia*, Bernhard Frick, Manuela Nania, Andrew G. Livingston and João T. Cabral*

December 11, 2021

We are grateful to all reviewers for their constructive and perceptive comments on our manuscript. We have revised the manuscript following the additional, extensive recommendations of Reviewer 2, and marked **track changes in red** in the revised manuscript. Below we provide a point-by-point reply to all comments.

Reviewer 1

I reviewed this paper before and still have the same reservations regarding its suitability for a general science venue. We understand the reviewer's reservation and have now revised the text and presentation of results in a more accessible and clearer way.

Reviewer 3

The manuscript provides some new and interesting results related to the confined water dynamics in polyamide RO membranes. The work is supported by some of the latest advanced membrane characterization studies and may facilitate deep understanding of the transport mechanisms involved in RO membranes. The authors have properly addressed all my review comments (provided through Nature Nanotechnology). Therefore, I recommend the publication of the manuscript. We thank the reviewer for their positive assessment of the manuscript, following the detailed revisions based on their input.

Reviewer 2

Several points raised by the reviewer were answered by the authors who provided useful and convincing new information. Some figures were also re-elaborated to be clearer, which is also appreciated. However, the reviewer believes that the data analysis is still lacking some precise descriptions and, therefore, the conclusions are not fully supported in a comprehensive manner. Only by providing the requested details and answers can the work be publishable on Nat. Com. We are grateful for the positive comment on the improvements made following the previous round of reviews. We were keen to address the extensive, additional comments raised which, in our opinion, are constructive and benefit the manuscript's clarity and rigour.

The analysis relies on the detailed analysis of two sets of data, namely BS (backscattering, IN16B) and Time-of-Flight (TOF, IN5). The resolutions are very distinct, therefore “slow” motions are probed by BS while fast motions are probed on IN5. Nevertheless, the “slow” motions are present in the elastic line on IN5, while the fast motions are present in the background of the BS measurements. The authors decided to perform a decoupled analysis, which means that they do not have one single analytical function that they fit to the whole set of data, but they assume that they have a double component dynamics (Eq 3 ; Eq 13 of the SI): one term accounting for polymer dynamics (Lorentzian 1), and one term accounting for water dynamics (Lorentzian 2). The respective amount of

these terms is weighted by using the volume fraction of polymer and water. For each species, then, a fraction of mobile and immobile protons is introduced to balance the elastic and quasielastic contributions. This approach can be acceptable, although, of course, it is a simplified frame that raise some questions, in particular relative to the claim of a multimodal dynamics, which is needed to be clarified by stating the underlying hypothesis. We have taken on board the recommendation and have adopted a single analytical function to describe all data, since the same protons are involved in the various motions. The expression is of course more complex since it considers (in the fully hydrogenous contrast) both polymer and water (mobile and immobile) protons. We nevertheless assume decoupling of polymer and water motion, since this simple model describes the data. Given the decoupled timescales of the dynamic terms probed in TOF and BS, our results remain largely unchanged for the diffusion/relaxation timescales, but the associated populations and their relation is now much clearer. We are glad that the reviewer insisted on this point, which does improve the clarity of the work.

My main concerns are three-fold (detailed comments are provided below):

- Fig. 5 is misleading. The nature of proton motions and corresponding populations must be better introduced and explained in the text, and sketched differently. We have now revised the schematic in Figure 5 (page 7) and provided information regarding the populations and their nature, based on the 'unified' model for analysis.
- The nature of slow motions is not unambiguously supported by the data. The text introducing this analysis is confusing, fitting parameters are missing, the reviewer asks some clarifications. We have expanded on the description of the unified model and have added data for the relevant populations associated to the various motions in Figs. 3 and 4.
- Multi-modal dynamics is not clear. What does it mean? Does it refer to polymer vs water? Does it refer to interfacial vs bulk? Does it refer to slow vs fast dynamics? Does it refer to motions in pores or through polymer? The various possibilities must be defined by the authors as a preamble of their paper, in a (missing) paragraph where they explain their assumptions and methodologies, and it must be highlighted again in the conclusions. We have now clarified the model assumptions and associated physical meaning. Given the experimental and computational evidence for PA membrane heterogeneity, we associate the multimodal dynamics of water to 'pore' and 'polymer network' diffusion. Other interpretations cannot be ruled out, however this physical picture is coherent with structural and MD data, and yields reasonable transport predictions, validated by data. As suggested we have expanded the introduction of such models.

Detailed comments to be addressed:

PRINCIPLE OF THE QENS ANALYSIS – QUESTIONS ON MULTI-MODAL DYNAMICS

A - In the dry and D₂O-hydrated samples, the authors show that the polymer dynamics is the only constituent they need to reproduce the data. The reviewer is now convinced by this result. Hence, the polymer component is determined in the H₂O-hydrated data and used as a fixed input to better evaluate the water dynamics. Exactly right. This procedure is required to isolate the water dynamics, and otherwise we could not obtain single valued data fits.

B - Most of the H₂O-hydrated results rely on the analysis of the TOF data. The HWHM dependence of the water dynamics Lorentzian (component 2) is analysed in the frame of both the confinement in a sphere model, yielding the confined diffusion coefficient D_{loc} (analysis of low-Q part of the HWHM profile, i.e. a plateau obtained at low temperatures) and the jump diffusion model (deviation from Fick's law at high Q, that allows to determine "another" diffusion coefficient called D_{tr} , as well as typical time τ_0 and length l , the latter being completely correlated to the two previous ones). This analysis, as raised in the first reviewer's report, is not optimized (using two models to describe the same HWHM dependence obtained by a single-Lorentzian fit is an approximation) but can be acceptable, keeping in mind that it provides information of the same protons/water molecules "seen" at different scales (low and high-Q parts of the profile). Therefore, D_{loc} and D_{tr} MUST be associated to the same type of motions and same type of translational diffusion. At this point, therefore, the scheme shown on Figure 5 is not consistent. The reviewer suggests to modify it and make clear that D_{loc} corresponds to the fast diffusion of a proton inside a confining sphere (as drawn), while D_{tr} corresponds to the jump diffusion of the same proton. The concentric circles added by the authors on Fig.5 are not understandable based on their assumptions. The "proton jumps" correspond to the

elementary step of the long-range diffusion mechanism that is evidenced here (because data are treated in the frame of a jump diffusion model), and this elementary process is ALSO THE ELEMENTARY STEP of the confined motions inside the sphere, by construction (one single- Lorentzian, one single process). To summarize:

1) decoupling the “concentric circles” motion from the “confined sphere” in Fig. 5 is misleading. We have now revised the schematic to illustrate the ‘confined sphere’ (associated with D_{loc}) and the same molecules diffusing across the ‘bulk’ polymer medium (D_{tr}) and finally the long range motion (D_{lr} , which we associate with the slower motion through dynamic pores. Within this framework, we have further computed the permeance (illustrated in SI, Fig S17 page 25) by weighing the contributions (D_{tr} and D_{lr}) by their respective partition coefficient (described in the main paper; page 7, line 498 onwards

2) one type of motion is evidenced from the TOF data (one fraction of mobile protons, one characteristic time τ_0 , one elementary jump length l , two diffusion coefficients because the same dynamics are considered at different length-scales (lowQ and highQ) : motions are fast in the 1nm sphere, but slowed down when the proton is travelling on longer distances ($D_{tr} < D_{loc}$). This is why the reviewer, in his first report, raised the point of “sub-diffusive” behavior, or, said in a different manner, a space-dependent distribution of diffusion coefficients. This point should be at least mentioned by the authors. By the way, this is somehow reflected in the final graph where the authors plotted their diffusion coefficients together with the ones from literature, in particular MD. The spread in values from MD is exactly reflecting the sub-diffusivity... The reviewer is not asking to re-do any analysis in a sub-diffusive analytical framework but to mention this aspect when they comment these values. In fact, the reviewer is convinced that multi-modal is not adapted to describe the complex behavior in these membranes and rather multi-scale, or scale-dependent, which, at least, puts all data in a bigger picture based on a physical mechanism that is well-known in situations of harsh confinement. The reviewer also suggests that the translational diffusion coefficient D_{tr} may correspond to long-range diffusion between confinement domains, e.g. rather correspond to the “red line” motions drawn on the figure, which is associated to the BS data by the authors without strong justification (see point C). We appreciate the comments about space-dependent diffusion, and the possibility of employing a sub-diffusive framework to analyse the data. We have revised and expanded on this discussion on page 6 (line 395 onward) that accompany the ‘unified’ model of analysis and updated schematic.

C – Once the data from TOF were analysed, the authors analyse the BS data. They, again, use a twocomponent function, one Lorentzian for the polymer dynamics, one for the water dynamics. The latter is completely decoupled from the TOF analysis (it is in fact Lorentzian 3). It can be accepted, if clearly stated. The need for 2 components is demonstrated in Fig. 12 of the SI and is convincing (of course, a background is needed, that would be the TOF water signal). However, the determination of the diffusive nature of the observed slow water motions is not convincing (Figure.SI-12, panels q-r). The polymer Lorentzian is an almost flat HWHM (blue and purple, q), with some small deviations at small Q that can be neglected because it comes close to the resolution and fits are difficult (what about the background variations with Q ??). However, the HWHM profiles obtained from the water component (red) are somehow very similar. The reviewer guesses that the dashed black line in (r) is the fit using a jump diffusion model (not described in the legend), from which values of the third diffusion coefficient labeled D_{lr} are extracted. This profile is resembling the one from the polymer (just shifted to higher HWHM, faster motions). The diffusive behavior is questionable. It could well be a localized motion, with not much reliability given on the first 2-3 low Q-values results. What are the values of the mobile fraction of protons and the elementary times related to this slow jump diffusion mechanism ? What are the associated jump lengths and how do they compare to TOF values of l ? Comparison between the fraction of BS mobile water protons with TOF mobile water protons is mandatory to have insights into the amount of water involved in this second type of dynamics. The question is: are these protons the same moving much slower, or are these protons located elsewhere and not exchangeable within the timescale ?? By answering this, one can then correctly refine the scheme of Fig. 5. and seriously claim about decoupled dynamics, multi-modal dynamics, and link with MD findings. It could be that there are several paths for diffusion in the material due to the hierarchy of pores and scales, but, as it is, the authors discussions are only speculative.

As mentioned above, in the revised manuscript, we have reformulated our analysis in terms of a single unified $S(q, \omega)$ function, for both TOF and BS data, accounting for both the polymer and the water contributions. We have therefore considerably revised this section of the paper, as well as the SI extensively. When fitting the TOF and BS data to these expressions, we show that the signals for the faster and slower motions become effectively an apparent broad background, or a narrow elastic line. While the key findings of our work do not change, we agree that this is a more rigorous (and elegant) way of presenting the study; moreover, it provides a more self-consistent analysis of the relative populations of mobile and immobile protons within specific time windows and type of motion. In order to corroborate our methodology, we have also carried out selected analyses in the time domain, by fitting the intermediate structure factor (illustrated in SI, Fig S14 page 19). Although the numerical results are not altered within uncertainty, this re-analysis strengthens our interpretation, and is simpler to the readership. We disagree however that D_{tr} could be incorrect and experimental data by several authors find values commensurate with those reported here (referenced in Fig. 5). However, we do agree that all motions could potentially be coupled, both the various diffusion modes for water, as well as those of water diffusion and polymer relaxation. We emphasise that PA membranes are spatially heterogeneous and characterised by aggregate and network pores which thus lead to (at least) bimodal dynamics, as suggested by the majority of simulations (cited in the main paper). This was the rationale for our decoupled analysis. We therefore make this reasonable assumption in the paper, which account for all experimental data. However, we do agree that this analysis framework (as detailed above) may not be the only framework able to describe the data (notably some coupling of polymer relaxation and water diffusion cannot be ruled out, and certainly water is likely to undergo rotational relaxations that we have not included in our model, to avoid overfitting the data); we have therefore expanded on this discussion (main paper; page 4, line 265 onward), and comprehensively revised Supplementary text 6, describing the model.

Globally, the reviewer asks for a better description of the methodology (specially for non QENS specialists, such that the findings are accessible to a broad audience): We have followed the reviewer's recommendation in the revised model description. We have also followed in detail the recommendations regarding the narrative and the introduction of the model, underlying assumptions, and physical picture. We agree that there remains a level of ambiguity relating to the origin of multimodal dynamics. While the correspondence with the engineering performance data seems compelling, we believe that additional MD simulations anchored by our findings will be needed to refine the physical picture. We expect that a distribution of diffusion coefficients might likely be needed to fully describe the complex water dynamics in a spatially heterogeneous and confined medium, and that the three coefficients that we experimentally resolve might correspond to principal components of such distribution. We have revised the discussion on this important, conceptual point.

Few additional comments: line 12 : a generally
line 372 : as well as

We thank the referee and we have fixed the typos.

Supporting info : Figure S2. Not clear at all. Polymer contribution is not indicated, while it is surely one of the Lorentzian components. Q value is not given. Local diffusion (Dloc) and translational diffusion (Dtr) is the same Lorentzian, therefore not well indicated by the arrows. So-called long-range diffusion (Dir) should correspond to the slow dynamics ??

Figure S2 only intends to provide a general overview, for non-experts, of how using different instrumental resolutions (and neutron spectrometers) it is possible to disentangle complex dynamics.

Figure S8. a-c : not $S(q, \omega)$ but $S(\omega)$ (integrated over Q).

Correct typo in legend : ESIF (EISF)

Figure S9. for both the dry...

p14 : remain remain

Figure S10. Specify what is the component (polymer) and measurement (IN16B).

Figure S12. Specify what is the dashed black line in (r) and refer to the equation used and parameters extracted from it.

Figure S13. Specify the type of measurement (TOF) from which data in a-f are extracted.

We thank the reviewer for their careful examination of the paper and have fixed all these points.

REVIEWERS' COMMENTS

Reviewer #2 (Remarks to the Author):

The authors have carefully revised their manuscript taking into account all my comments and significantly developing the key points regarding QENS data analysis using a complex diffusion process modelling, which they have clarified and consistently described in the new version of their work. Therefore, the reviewer is satisfied and finds the paper suitable for publication.

Draft Only